# Reliance upon ancestral mutations is maintained in colorectal cancers that heterogeneously evolve during targeted therapies

Mariangela Russo[1], Simona Lamba[1], Annalisa Lorenzato[1,2], Alberto Sogari[2,3], Giorgio Corti [1], Giuseppe Rospo[1], Benedetta Mussolin[1], Monica Montone[1], Luca Lazzari[1,2], Sabrina Arena [1,2], Daniele Oddo[1,2], Michael Linnebacher[4], Andrea Sartore-Bianchi[5,6], Filippo Pietrantonio[6,7], Salvatore Siena[5,6], Federica Di Nicolantonio [1,2] & Alberto Bardelli[1,2]

Attempts at eradicating metastatic cancers with targeted therapies are limited by the emergence of resistant subclones bearing heterogeneous (epi)genetic changes. We used colorectal cancer (CRC) to test the hypothesis that interfering with an ancestral oncogenic event shared by all the malignant cells (such as WNT pathway alterations) could override heterogeneous mechanisms of acquired drug resistance. Here, we report that in CRC-resistant cell populations, phylogenetic analysis uncovers a complex subclonal architecture, indicating parallel evolution of multiple independent cellular lineages. Functional and pharmacological modulation of WNT signalling induces cell death in CRC preclinical models from patients that relapsed during the treatment, regardless of the drug type or resistance mechanisms. Concomitant blockade of WNT and MAPK signalling restrains the emergence of drug-resistant clones. Reliance upon the WNT–APC pathway is preserved throughout the branched genomic drift associated with emergence of treatment relapse, thus offering the possibility of a common therapeutic strategy to overcome secondary drug resistance.

[1] Candiolo Cancer Institute-FPO, IRCCS, 10060 Candiolo, Turin, Italy. [2] Department of Oncology, University of Torino, SP 142 km 3.95, 10060 Candiolo, Turin, Italy. [3] FIRC Institute of Molecular Oncology (IFOM), 20139 Milan, Italy. [4] Department of General Surgery, University of Rostock, Rostock D-18057, Germany. [5] Niguarda Cancer Center, Grande Ospedale Metropolitano Niguarda, Milan 20162, Italy. [6] Department of Oncology and Hemato-Oncology, Università degli Studi di Milano, Milan 20122, Italy. [7] Medical Oncology Department, Fondazione IRCCS Istituto Nazionale dei Tumouri, Milan 20133, Italy. Correspondence and requests for materials should be addressed to M.R. (email: mariangela.russo@ircc.it) or to A.B. (email: alberto.bardelli@unito.it)

Pharmacological blockade of oncogenic mutations (such as EGFR or BRAF alterations) has not only shown clinical effectiveness in advanced colorectal cancer (CRC), but also in melanoma, lung and other tumour types[1]. Unfortunately, clinical response is often transitory and almost all patients succumb to the disease due to acquired drug resistance. Preclinical studies have shown that blockade of oncogenic signalling with targeted agents may lead to the clonal expansion of pre-existing low frequency cell clones carrying alterations conferring drug resistance, which eventually become dominant in the population leading to treatment failure[2–4]. We and others have previously found that resistance mechanisms to agents blocking oncogenic proteins can be molecularly heterogeneous, and often include genetic alterations in downstream effectors of the same pathway, and/or activation of parallel bypass pathways[3,5–7]. This phenomenon has also been observed in patients, whereby individual metastatic lesions were shown to independently evolve distinct resistance mechanisms, which translated into lesion-specific response to subsequent lines of therapy and consequent clinical failure[8,9]. It has been proven extremely difficult to engage with subsequent lines of therapy the heterogeneous mechanisms of resistance and the subclonal pattern of tumour cell populations that emerge upon drug selection[8]. CRC displays molecular heterogeneity during tumourigenesis and therapeutic treatment[10–13]. In analogy with the structure of the trees, *trunk* mutations represent the complement of genetic alterations that occur in first cell division during tumour development, thus being present in all malignant cells (clonal mutations). All mutations that occur after the most recent appearance of a common ancestor are instead subclonal (branched mutations)[14,15].

We reasoned that molecular determinants shared by every cell subclone (*trunk*) might be better suited as therapeutic targets than heterogeneous events in the branches, as the former remain present in each drug-resistant cell independently from its genetic drift. In line with this, WNT/β-catenin signalling in CRC is a paradigmatic example of cancer *trunk* pathway, as mutations affecting its molecular switches occur at the adenoma stage and are present in all cells when the disease becomes metastatic[16,17].

The adenomatous polyposis coli (*APC*) gene is a key negative regulator of the canonical WNT signalling pathway, by providing a scaffold for the destruction complex that stimulates phosphorylation and subsequent ubiquitin-dependent degradation of β-catenin. Loss of function (LOF) mutations in the *APC* gene or gain of function (GOF) mutations in the *CTNNB1* gene (encoding for β-catenin protein) are found in more than 80% of the sporadic CRCs[16,18–21].

Most of cancer-linked *APC* variants are nonsense mutations, occurring in the mutation cluster region resulting in premature stop codons and a truncated gene product lacking the carboxy-terminus of the protein[20,22]. Because these truncations cause loss of the domains required for binding to β-catenin, APC inactivation leads to accumulation of nuclear β-catenin, which in turn activates the WNT signalling target transcription factors (T-cell factor or Tcf) and the lymphoid enhancer factor (LEF)[23], resulting in hyperactivation of the pathway.

In addition to APC and β-catenin, the E3 ubiquitin ligases ring-finger protein 43 (RNF43), and zinc and ring finger 3 (ZNRF3) also negatively regulate WNT signalling by promoting ubiquitination and subsequent degradation of the Frizzled and LRP5/6 WNT pathway receptors[24,25]. The secreted WNT agonists of the R-spondin family, RSPO1-4, in turn, negatively regulate RNF43/ZNRF3. LOF mutations of *RNF43/ZNRF3* genes[26,27] and GOF gene fusions involving *RSPO2* and *RSPO3*[20,28] lead to increased cell surface abundance of WNT receptors and consequently constitutive activation of WNT signalling in the 15–20% of CRC that lack *APC* or *CTNNB1* alterations[20,21].

CRC cells are known to rely on constitutively active WNT/β-catenin signalling, since restoration of wild-type (WT) APC function affects their proliferation[29] and can suppress their tumourigenicity[30].

On the other hand, CRC displays molecular heterogeneity[10–13]; whether and to what extent CRCs, developing subclonal distinct molecular lineages as a result of the drug treatment, remain dependent on the truncal WNT signalling hyperactivation is largely unknown.

We report that the functional and pharmacological modulation of WNT signalling in CRC cells and patient-derived models restricts cell growth and leads to cell death, despite multiple pro-survival mechanisms acquired previously under treatment with clinically relevant targeted agents. We further find that concomitant blockade of the MAPK and WNT pathways restrains clonal evolution, and prevents the onset of resistance.

## Results

**Treatment with targeted agents fuels molecular heterogeneity.** To test whether dependency on WNT signalling was maintained in CRC cell populations that developed multiple heterogeneous mechanisms of targeted drug resistance, we first generated populations of cells resistant to the BRAF inhibitor dabrafenib, alone or in combination with the anti-EGFR monoclonal antibody cetuximab (Supplementary Fig. 1a, b; Supplementary Table 1), as combination regimens have shown promising activity in BRAF-mutated metastatic colorectal cancer (mCRC) patients[31]. Whenever possible, multiple independent resistant models for each cell line were obtained. To extend our findings beyond BRAF-mutant CRC, we also characterised a previously established collection of RAS/BRAF WT cell lines, which were made resistant to the blockade of oncogenic kinases including anti EGFR antibodies and the NTRK inhibitor entrectinib (Fig. 1 and Supplementary Table 1)[2,3,5–7,32].

Molecular profiling of resistant cells unveiled that in most instances several, often concomitant, mechanisms of drug resistance emerged affecting either the drug target (such as secondary mutations in *EGFR* or *NTRK1*) or effectors in the same

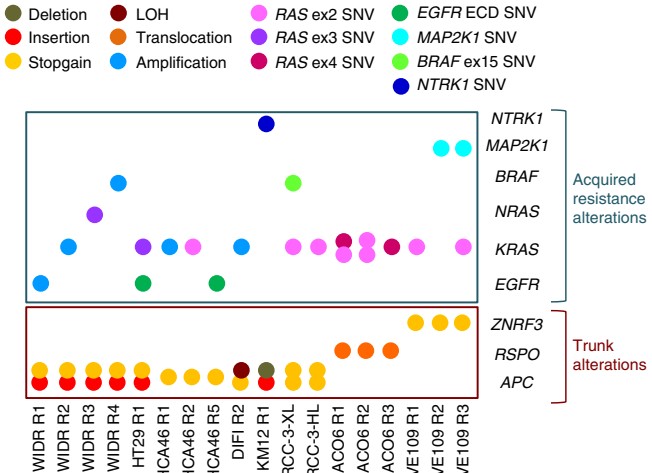

**Fig. 1** Heterogeneous mechanisms of secondary resistance to targeted therapies in colorectal cancer (CRC) cells. Indicated CRC cells were made resistant to single targeted agents or combination of them (see Supplementary Table 1). Trunk alterations in the WNT pathway are depicted in the lower brown box. The upper blue box illustrates multiple, often co-occurring, genetic alterations acquired at secondary resistance. SNV indicates Single Nucleotide Variance. Ex stands for exon. ECD stands for extra-cellular domain

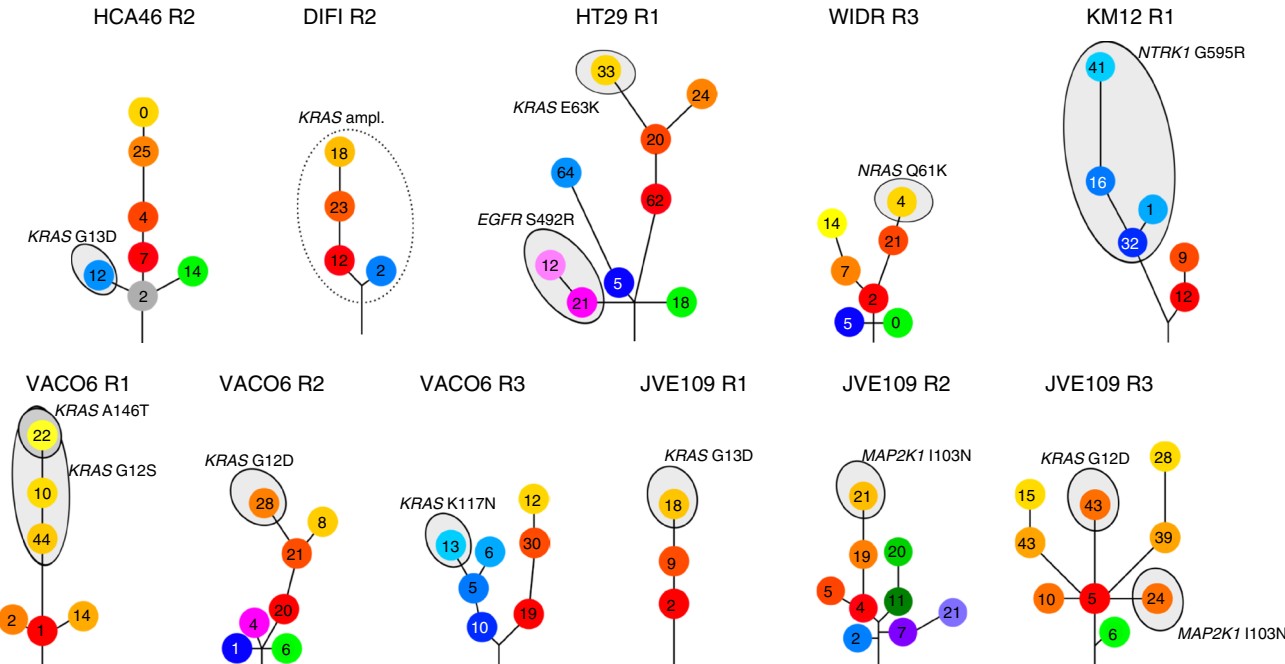

**Fig. 2** Clonal evolution of CRC cell populations upon secondary resistance to targeted agents. Phylogenetic evolutionary maps illustrate the development of sub clonal populations after acquisition of secondary resistance to the targeted therapies. The bioinformatic tool EXPANDS was used to infer the clonal architectures using gene copy number, synonymous and non-synonymous somatic mutations, as described in detail in the Material and Methods section. Each circle represents a subclonal population, numbers indicate non-synonymous variations defining clonal sweeps. Length of the branches is proportional to the number of variants (synonymous and non-synonymous) acquired by individual clones, while ancestral branches define the main colour of its subclones. Subpopulations carrying somatic alterations known to drive drug resistance are highlighted (see Supplementary Table 1). The dashed line indicates KRAS amplification

or in parallel pathways (Fig. 1). Trunk genomic alterations in the WNT/β-catenin pathway were maintained in resistant cell populations. These changes included stop codon and indel mutations, leading to a premature C-terminus of APC protein or molecular alterations in upstream components of the WNT pathway (Fig. 1 and Supplementary Table 1).

**Phylogenetic subclonal structures of CRC resistant cells.** Exome analyses revealed that—beyond putative key driver events in oncogenic kinase signalling responsible for drug resistance—several novel genetic alterations were acquired following selective pressure of the targeted agents. Of these, some were shared (common), while most were "private", suggesting parallel independent patterns of evolution under drug-induced selective pressure (Supplementary Fig. 1c, d). We applied bioinformatic tools to the exome data to infer the clonal architecture of each resistant population. Using clone phylogenetic tracking, we found that resistant cell populations displayed complex subclonal architecture, indicating concomitant evolution of multiple cellular lineages during treatment, each associated with specific sets of molecular alterations (gene copy number, synonymous and non-synonymous somatic alterations) (Fig. 2). This phenomenon occurred independently from the type or the number of drugs applied to achieve resistance (Supplementary Table 1).

While some of the evolutionary branches displayed well-known resistant mutations (such as RAS, MAP2K1 and EGFR extracellular mutations), others did not (Fig. 2). This is in agreement with mutant allele frequencies determined by exome analysis (Supplementary Table 2) and suggests that additional mechanisms of drug escape remain to be characterised.

To verify the mutation's co-occurrence/exclusivity patterns predicted by phylogenetic tracking, we performed single cell dilution of the resistant populations. Droplet digital PCR (ddPCR) analysis of the individual clones isolated from HT29 R1 revealed either EGFR

p.S492R or KRAS p.E63K mutations (Supplementary Table 3). In clones isolated from JVE109 R3, we detected KRAS p.G12D or the MAP2K1 p.I103N variants (Supplementary Table 3), confirming the patterns inferred by bioinformatic analysis. Occasionally we found wells containing both mutations (Supplementary Tables 3), in these instances, the different fractional abundance suggested the presence of a mixed population of cells likely associated with imperfect single cloning procedures (Supplementary Table 4).

**Restoration of WT APC overcomes acquired drug resistance.** We sought to investigate whether colorectal tumours that had developed subclonal distinct molecular lineages as a result of drug treatment remained dependent on WNT signalling. At the present time, APC is not directly druggable, and approaches aimed at targeting the upstream components of the WNT/β-catenin pathway would have minimal effect in APC-mutant cancers. As a proof of the concept strategy, we therefore decided to ectopically reintroduce WT APC in CRC cells carrying APC-inactivating mutations. Restoration of functional WNT signalling impaired growth of both parental and derivative resistant APC-defective cells, leading to rapid cell death 48 h after APC nucleoporation (Fig. 3; Supplementary Fig. 2). By contrast, ectopic expression of a truncating inactive form of APC (p.G97*) only marginally affected the cell growth. Notably, WNT signalling restoration inhibited the growth of drug-resistant cells independently of the molecular mechanisms of resistance or the oncogenes and pathways involved in drug escape (Supplementary Fig. 2).

**Inhibition of WNT signalling in drug-resistant CRC cells.** While the development of therapeutic strategies directly targeting APC remains challenging, other key nodes of the WNT pathway in CRCs may be amenable to pharmacological approaches[33]. For instance, inhibition of porcupine (PORCN), an acyltransferase

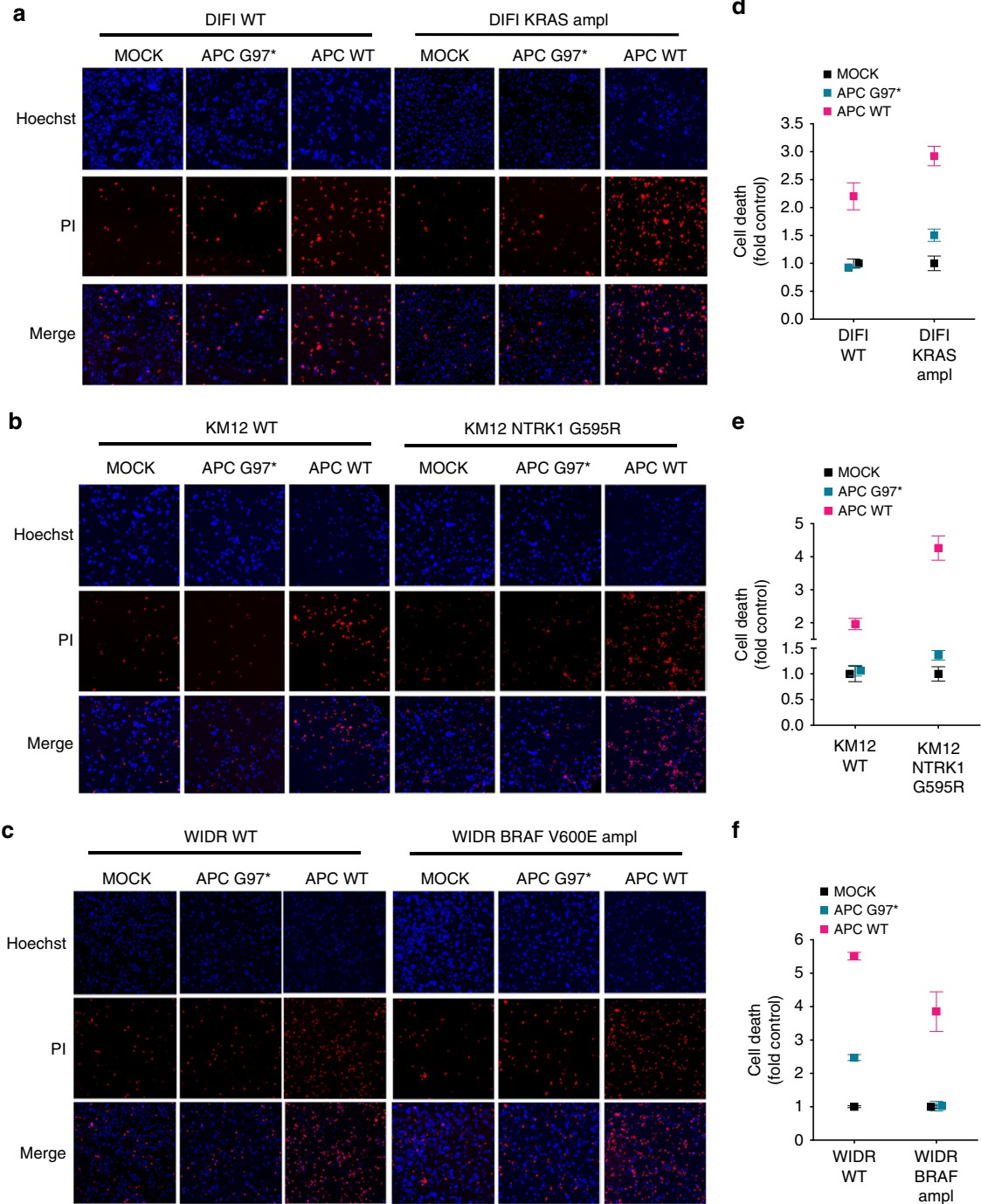

**Fig. 3** Functional restoration of WTAPC induces cell death in CRC cells carrying distinct mechanisms of secondary resistance. **a–c** Parental and resistant-derivatives CRC cells were electroporated with plasmid encoding for WTAPC or an inactive APC version (G97*). Electroporation buffer alone was used as control (mock). After 48 h, cells were stained with Hoechst 3342 /Propidium Iodide (PI) to detect cell death. Representative images of single 96-wells are shown for each condition. **d–f** Relative quantification of Hoechst/PI positive cells was made using ImageJ software and normalised against mock cells. Results represent means ± SD of three independent wells

required for intracellular transport, secretion, and activity of WNT ligands, has been remarkably effective in CRCs carrying *RSPO2/3* re-arrangements or *RNF43/ZNRF3* truncating mutations[34,35]. To test the impact of modulating the WNT signalling pathway in CRC after failure of targeted therapies, we used LGK974, a clinical-stage (NCT01351103) porcupine inhibitor[36]. We identified three CRC cell lines harbouring trunk alterations in *RSPO3*[35] or *ZNRF3* genes with exquisite sensitivity to LGK974 (Supplementary Fig. 3a, b) and which lack mutations in *APC* and

*CTNNB1* (β-catenin) (Fig. 1). We then measured the β-catenin-dependent transcriptional activity of Tcf/LEF transcription factors. Treatment with LGK974 severely reduced the β-catenin activity in *RSPO3/ZNRF3* altered cells (Supplementary Fig. 3c). In CRC cells, AXIN2 is transcriptionally induced following reception of a WNT/β-catenin signal, and represents a marker of WNT pathway functionality[30,37]. Porcupine inhibition promoted on-target gene modulation, as shown by reduced expression of *AXIN2* (Supplementary Fig. 3d).

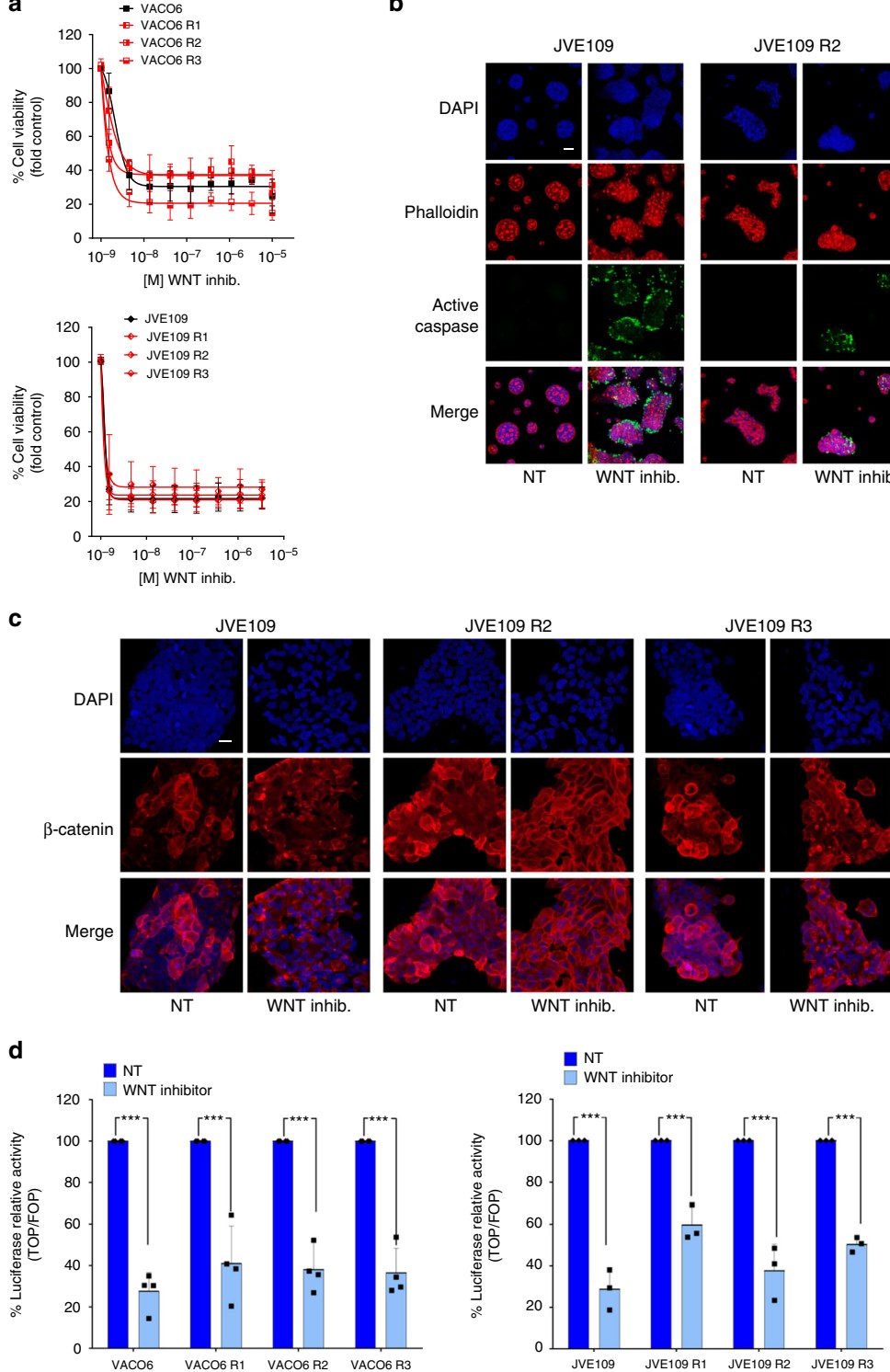

**Fig. 4** Pharmacological blockade of WNT signalling is effective in molecularly heterogeneous populations of drug-resistant CRC cells. **a** BRAF-mutated VACO6 and JVE109 CRC cells were treated for 5 days with increasing concentrations of porcupine inhibitor LGK974 (WNT inhib.). Cell viability was assayed by the ATP assay. Data points represent means ± SD of at least three independent experiments. **b** JVE109 parental and resistant-derivatives cells were treated with LGK974 for 5 days. After that, active cleaved caspase-3 was detected by immunofluorescence (green). Nuclei are stained with DAPI (blue) and actin with Phalloidin (red). Scale bar: 50 μm. **c** JVE109 parental and resistant-derivatives cells were treated with LGK974 for 4 days. Representative confocal microscopy images showing β-catenin distribution (red) are reported. Nuclei are stained with DAPI (blue). Scale bar: 25 μm. **d** WNT inhibitor LGK974 induces a strong downregulation of β-catenin-dependent transcriptional activity of Tcf/LEF luciferase reporter construct in CRC parental and resistant-derivatives cells. Results represent means ± SD of at least two independent experiments. Single points indicate results of single experiments. ***$p < 0.001$ (Student's $t$ test)

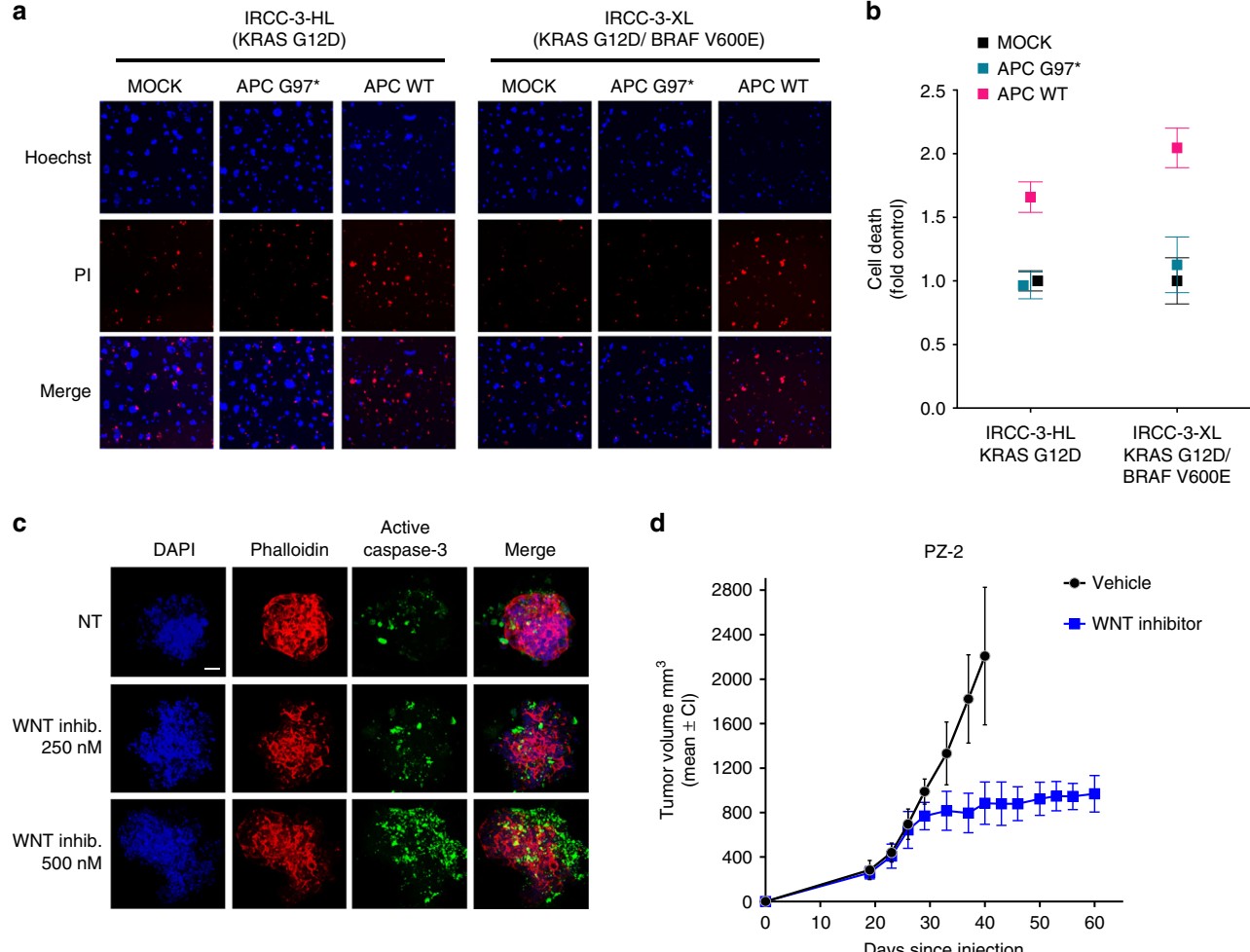

**Fig. 5** Cells and organoids from drug-resistant CRC patients rely on the WNT/β-catenin pathway. **a** Primary 2D cell lines, established from the tissue specimen collected from a CRC patient whose tumour developed secondary resistance to anti-EGFR therapy (see detailed methods section), were electroporated with a plasmid encoding for WT APC or an inactive APC version (G97*). Electroporation buffer was used as the control (mock). After 48 h, cells were stained with Hoechst 3342 /Propidium Iodide (PI) to detect cell death. A representative image of a single 96-well is shown for each condition. **b** Relative quantification of Hoechst/PI positive cells was made using ImageJ software and normalised against mock cells. Results represent means ± SD of three independent wells. **c** 3D organoids established from a CRC patient whose tumour developed secondary resistance to EGFR-BRAF combinatorial treatment (see detailed methods section), were treated with LGK974 for 2 weeks. Representative confocal microscopy images showing active cleaved caspase-3 (green) are shown. Nuclei are stained with DAPI (blue) and actin with Phalloidin (red). Maximum projection of a 10 image stack along the z-axis. Scale bar: 50 μm. **d** Patient-derived mice models (xenopatient) were established from a tumour obtained from a metastatic colorectal cancer patient (PZ-2) resistant to EGFR/BRAF combinatorial treatment. Upon successful engraftment, mice were randomised to vehicle ($n = 6$) or LGK974 (WNT inhibitor) ($n = 6$) treated arm. Results represent tumour mass volume (mm³, mean ± CI of individual tumour volume)

Derivative cell populations with heterogeneous secondary resistance alterations to MAPK pathway inhibition retained the same level of sensitivity of their parental counterparts to modulation of the WNT pathway. LGK974 impaired cell growth (Fig. 4a) and promoted cell death through caspase 3/7 activation in a dose-dependent manner in resistant cells, regardless of the molecular evolution, which occurred during previous target drug exposure (Fig. 4b; Supplementary Fig. 4).

To further assess the molecular mechanisms of action of LGK974 in drug resistant cells, both distal and proximal WNT signalling events were examined. We found that LGK974 downregulated phosphorylation of the WNT co-receptor LRP6 and in parallel triggered the accumulation of Axin1 (Supplementary Fig. 5a), a member of the β-catenin destruction complex, which with APC promote, the ubiquitin-dependent proteasomal degradation of β-catenin via CK1α- and GSK3β-mediated phosphorylation of β-catenin[38–40]. Nuclear exclusion of β-catenin occurred both in

parental and resistant derivatives (Fig. 4c), and resulted in strong reduction of β-catenin-dependent Tcf/LEF transcriptional activity (Fig. 4d). The marked response to the inhibition of WNT ligands secretion was associated with a corresponding decrease in the expression of WNT target genes *AXIN2* and *LGR5* indifferently in parental and resistant derivatives (Supplementary Fig. 5b, c).

**Reliance upon WNT-APC pathway in patient-derived CRC models.** To extend the cell-based findings to more clinically relevant models, we exploited the patient-derived cancer cells and organoids, which we established from two patients with mCRC who initially responded, and then progressed upon treatment with the targeted therapies. A tissue biopsy was collected when a patient with an initial RAS/BRAF WT tumour developed secondary resistance to anti-EGFR-based therapy. Based on our previous experience and to improve chances of establishing

patient-derived models, the biopsy was divided in two fragments, one of which was used to generate a primary cell line (patient-derived cell line, -HL) (Fig. 5a), while the other was transplanted subcutaneously in an immunocompromised mouse. Upon successful engraftment of the latter, the tumour (PDX, patient-derived xenograft or xenopatient) was excised and employed to derive another primary cell line (PDX-derived cell line, -XL). Different (sub-clonal) mechanisms of resistance were identified in the two cell models. While both cell lines harboured the same *APC* ancestral mutations indicating a clonal origin (Supplementary Table 1), one displayed a *KRAS* p.G12D mutation, whereas the other showed a *BRAF* p.V600E variant (Fig. 1; Supplementary Table 1). Remarkably, ectopic restoration of the WT APC led to cell death in both patient-derived cell models, regardless of the resistance mechanisms that emerged in the tumour during clinical treatment (Fig. 5a,b; Supplementary Fig. 3).

A second biopsy was gathered from a mCRC patient whose tumour carried genetic alterations in *RNF43* and *BRAF* genes, and clinically responded and then relapsed to EGFR blockade with cetuximab in combination with the BRAF inhibitor encorafenib. The biopsy was first transplanted subcutaneously in an immunocompromised mouse (see Methods). After successful engraftment and growth, the tumour was excised and fragmented to generate cohorts of mice bearing patient-derived tumourgrafts (xenopatients), while one fragment was used to derive in vitro 3D organoids cultures. In these organoid models with acquired resistance to combinatorial EGFR and BRAF target inhibitors, inhibition of the WNT pathway by the porcupine inhibitor LGK974 promoted apoptosis in a dose-dependent manner, as indicated by caspase-3 staining (Fig. 5c). The patient-derived tumourgrafts generated from this BRAF-mutant tumour grew very rapidly, emphasising and reflecting the aggressiveness of

the malignancy (Fig. 5d black line) from which it originated. Due to this rapid growth rate, vehicle-treated mice had to be sacrificed 14 days after initiation of treatment, according to ethical guidelines. Nevertheless, inhibition of constitutively active WNT pathway markedly delayed tumour growth in the xenopatient cohort treated with porcupine inhibitor LGK974, inducing a prolonged tumour mass stabilisation (Fig. 5d blue line).

**WNT pathway modulation in CRC cells.** To characterise the efficiency of WNT inhibition, MAPK-resistant CRC cells were treated with the porcupine inhibitor LGK974 in a long-term assay. While progressive growth impairment was detected from day 5 to day 16 (Supplementary Fig. 6), in most cell lines, a slight increase in the cell viability was observed at day 21, suggesting the presence of 'persister' cells that might have survived the WNT pathway inhibition. Indeed, the induction of caspase activity declines after 3 weeks of treatment (Supplementary Fig. 6). Based on this finding, we wondered whether genetically defined sub-clones—identified prior to LGK974 treatment (Fig. 2)—might persist upon WNT pathway modulation. We therefore treated the CRC-resistant cells with WNT inhibitor for 2 weeks and then extracted gDNA from the cells that survived pharmacological treatment. ddPCR analysis unveiled that the fractional abundance of pre-existing mutant alleles remains substantially unchanged between the untreated and the LGK974-treated cells over 2 weeks (Fig. 6). Modest fluctuations of fractional abundance of mutated alleles were detected also in untreated cells in different biological replicates (Fig. 6), supporting the possibility that WNT-related cell death in CRC cells is largely independent from the oncogenic alterations they had acquired.

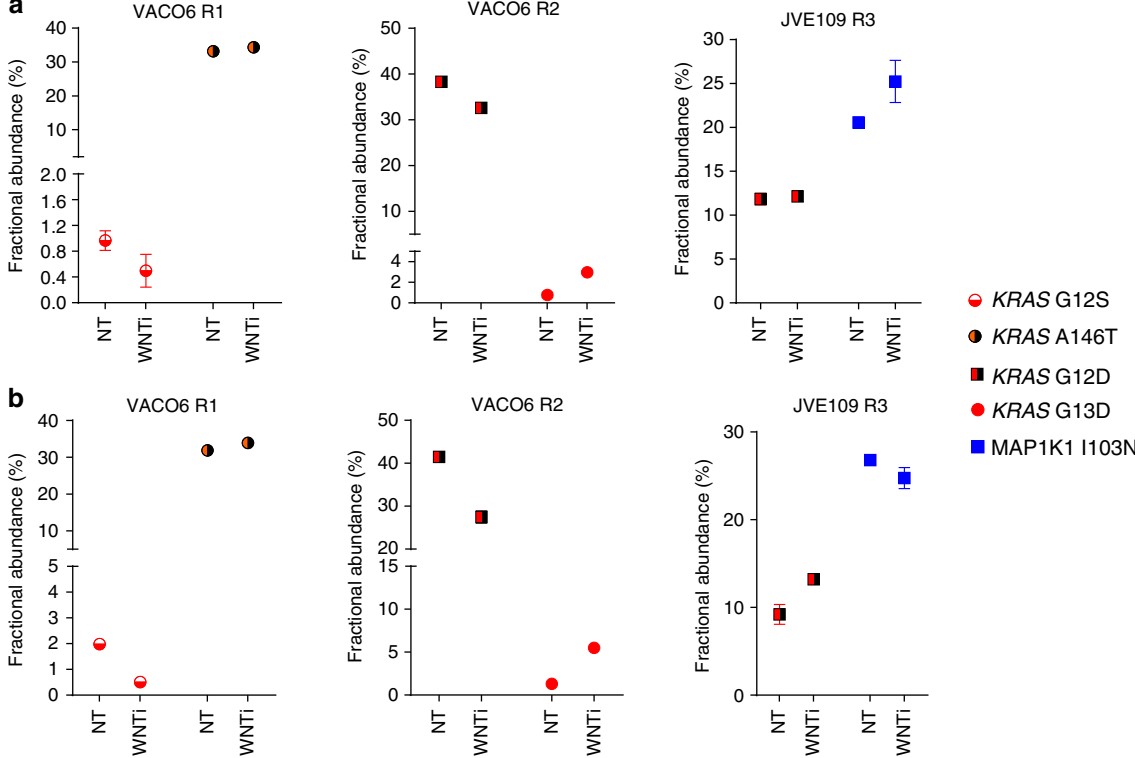

**Fig. 6** Cell death associated with porcupine blockade is independent from MAPK alterations. MAPK-resistant CRC cells were treated with 1 μM LGK974 (WNTi) for 2 weeks. After that, gDNA was extracted from the control untreated (NT) and the LKG974-treated cells. ddPCR analysis was performed to measure the fractional abundance of the mutated alleles, previously identified as mechanisms of secondary resistance to MAPK inhibition. Results represent means ± SD of two independent technical replicates. **a** and **b** indicate the independent biological replicates of the experiment

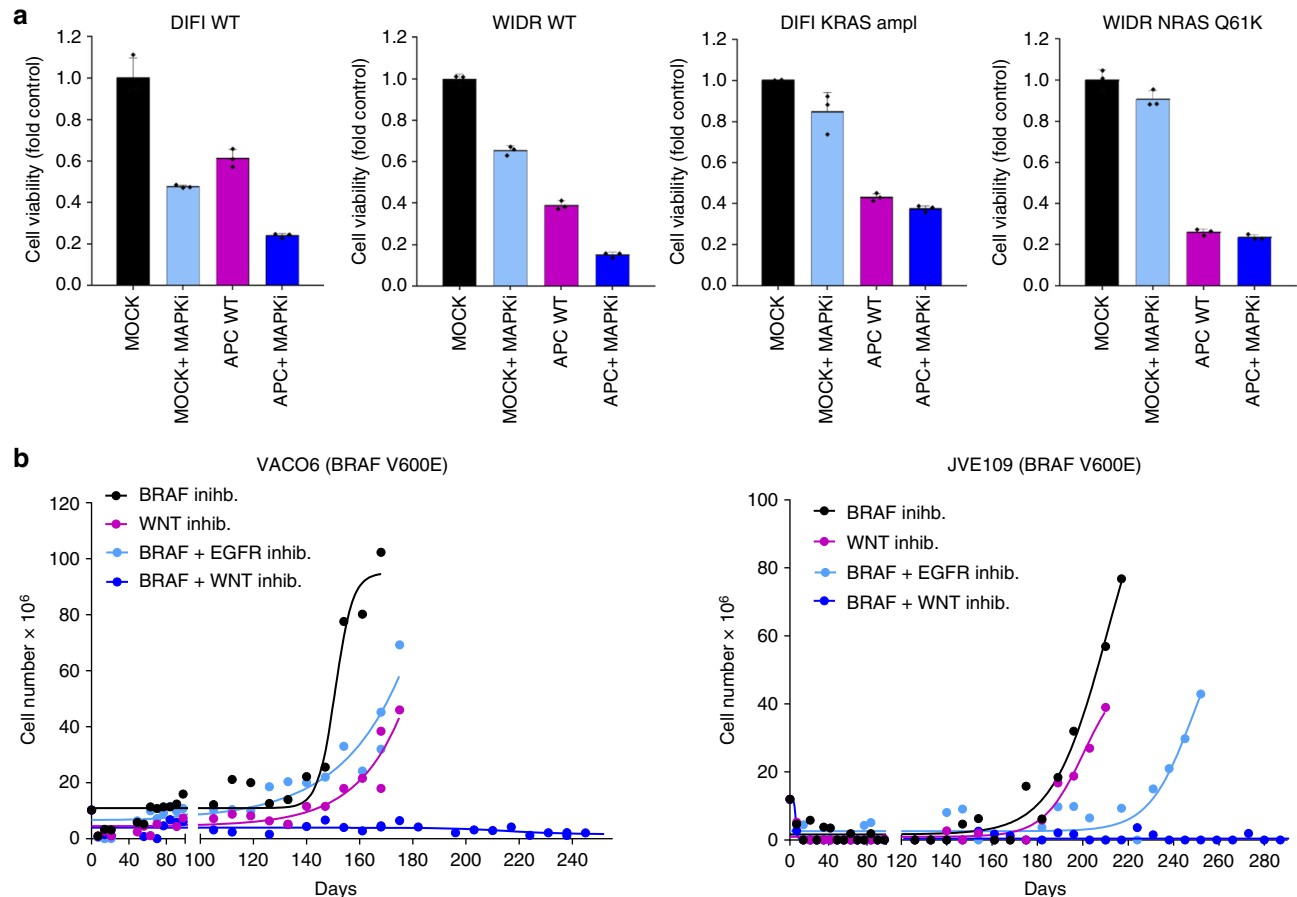

**Fig. 7** Concomitant blockade of WNT and MAPK signalling restricts the emergence of drug resistance. **a** APC-mutated CRC cells and their MAPK-resistant derivatives were transfected with plasmid expressing intact WT APC or control electroporation buffer (mock). After transfection, the cells were seeded in 48-wells plates with or without cetuximab (DIFI cells), dabrafenib + cetuximab (WIDR cells). After 48 h, cell viability was assayed by ATP assay. Representative graphs of two independent experiments for each cell line are reported. Results represent means ± SD of three independent wells. Single points indicate results of single experiments. **b** BRAF-mutated CRC cells were treated with dabrafenib (BRAFinhib.), dabrafenib + cetuximab (EGFRinhib.), LGK974 (WNTinhib.), or dabrafenib + LGK974, until secondary resistance emerged

**Inhibition of WNT-MAPK pathways prevents secondary resistance.** Tantalised by the above results, we tested whether combinatorial inhibition of WNT and MAPK pathways might be effective on cells that had already acquired resistance to targeted therapies. To test the hypothesis, we treated BRAF-mutant CRC cells made resistant to MAPK inhibitors with LGK974 alone or in combination with the BRAF inhibitor dabrafenib. In most of the cell models, BRAF inhibition reduced the effectiveness of WNT blockade on restricting growth (Supplementary Fig. 7). The only exception was JVE109 R2 in which horizontal inhibition of MAPK and WNT pathways induced a more efficient inhibitory effect. We noted that this is the only resistant population without RAS mutations (Supplementary Tables 1 and 2). We speculate that in all the other resistant models carrying concomitant BRAF V600E and RAS mutations, BRAF inhibition can paradoxically stimulate proliferation by promoting a known paradox biochemical activation of the MAPK pathway[41]. This may explain the partial rescue in cell viability we observed when dabrafenib is added to LGK974 in cells with acquired RAS resistance mutations.

In line with this, inhibition of MAPK pathway increases the efficacy of functional APC restoration in parental (MAPKi-sensitive) cell lines, while does not further enhance the impact of WNT pathway modulation when secondary resistance is already established (Fig. 7a).

Prompted by these results, we tested whether 'ab initio' (that is before onset of secondary resistance) blockade of the WNT and MAPK pathways could instead prevent or delay the evolution of resistant clones. To test this possibility, we performed an assay we previously developed to assess in preclinical models development of resistance over time, in analogy to the time to progression (TTP) value usually recorded in patients[42]. BRAF-mutant VACO6 and JVE109 cells were treated with the BRAF inhibitor dabrafenib or WNT inhibitor LGK974 alone, or in combination and the emergence of resistant subpopulations was monitored over time. This TTP assay showed that although inhibition of WNT pathway or MAPK pathway alone was initially effective, resistant clones emerged. On the contrary, concomitant suppression of WNT and MAPK signalling pathways prominently delayed the onset of relapse, with no resistant clones emerging even up to 9 months after treatment initiation (Fig. 7b). Remarkably, such a combinatorial approach was more effective than vertical dual inhibition of the MAPK pathway with BRAF and EGFR inhibitors, which is currently undergoing clinical evaluation (Fig. 7b).

## Discussion

The awareness that solid tumours are molecularly heterogeneous poses a formidable therapeutic challenge. We and others have previously shown, both in preclinical and clinical studies, that potentially aggressive subclones may be present at low frequency in the primary tumour and remain almost undetectable providing a heterogeneous reservoir to fuel resistance in response to

treatment selective pressures[2,4,13,43]. In addition, stressful conditions such as drug treatment can induce acquisition of novel mutations, as well as 'genome chaos', contributing to molecular heterogeneity[7,44].

The polyclonal landscape of CRC can result from several processes including: multi-step accumulation of genetic and epigenetic aberrations, alterations by Darwinian selection, neutral acquisition of passenger variants over prolonged time, and short periods of genomic instability, resulting in concomitant occurrence of several molecular changes[11,17,45–47]. The specific combinations of molecular alterations within a tumour thus affect not only the natural course of the disease, but also the clinical response to therapeutic regimens.

We find that the treatment with targeted therapies, although initially effective, fuels clonal evolution and further amplifies molecular diversity. Phylogenetic tracing of CRC populations that acquired drug resistance, unveiled the coexistence of numerous inter-mixed molecular lineages, each characterised by specific mutational signatures. Importantly, such complex sub-clonal architecture was observed not only in CRC cells treated with a single agent (such as EGFR inhibitor cetuximab or BRAF inhibitor dabrafenib), but also when combinatorial regimens of drugs targeting different pathways were administered.

In principle, deciphering the complete genomic profiles of each tumour would be crucial for precision medicine, in order to allow targeting of all genetically driver alterations concomitantly present in the tumour bulk. However, this remains difficult to achieve, as bioinformatic tools designed to infer phylogenetic tumour structures data are still being optimised.

Of note, exome analysis, although highly sensitive, did not reveal a readily recognisable mechanism of resistance in some of the subclonal populations highlighted by phylogenetic investigation. This is consistent with what is observed in patients. Treatments aimed at targeting acquired oncogenic nodes present in tumour branches, are active only on a subset of the tumour lesions, conceivably as a consequence of coexistence of multiple resistance mechanisms, some of which are often not molecularly defined or detectable. Indeed, the independent development of different resistance mechanisms in distinct metastases translates in lesion-specific response to subsequent lines of therapy and consequent clinical failure[8,9].

Therefore, what limits further progress in the field of targeted therapies is not the emergence of resistance -per se- but the fact that relapses are driven by parallel genomic evolution of multiple cell lineages, which become extremely difficult to eradicate.

Virtually all CRCs display aberrant WNT signalling as the initial tumourigenic event[17,20]. The impact of constitutive WNT pathway activation on colorectal tumourigenesis has been well characterised. It is known that restoration of APC function could revert an adenoma to normal colonic tissue[30], highlighting the importance of continuous WNT pathway activation for CRC maintenance. Much less is known as to whether reliance on the WNT pathway is preserved in later phases of colorectal carcinogenesis when tumours face genomic bottlenecks and evolution driven by administration of chemotherapy and targeted therapies.

Our results provide functional and pharmacological evidence that dependency upon deregulation of the WNT/APC/β-catenin signalling axis is maintained through the distinctive stages that characterise the emergence of resistant clones: cytotoxic bottleneck, clonal selection, adaptation, neutral evolution, acquisition of multiple molecular aberrations and expansion; thus offering broadly applicable therapeutic options to override heterogeneity.

We demonstrated that interference with WNT pathway hyperactivation through reintroduction of functional APC led to cell death in all the resistant CRC populations analysed, bypassing the multiple pro-survival mechanisms acquired under previous drug exposure. Notably, the cytotoxic effect of wild-type APC re-expression is rapid; suggesting that even partial inhibition of constitutively active trunk signalling could result in powerful anti-tumour effects.

Although restoration of wild-type APC function in CRC patients is currently therapeutically unfeasible, our results suggest that small molecules aimed at blocking constitutive WNT signalling at different levels, might achieve similar effects in defined patients subpopulations. In this regard, our findings indicate that CRCs harbouring trunk alterations in upstream components of the WNT pathway (such as RSPO3, RNF43 and ZNRF3) retain strong sensitivity to porcupine blockade despite acquisition of complex sub-clonal structure. Importantly, cell death secondary to inhibition of ancestral pathway hyperactivation occurs in the cell population, independently from the oncogenic alterations acquired under selective pressure of targeted agents. We cannot exclude that other mechanisms, for example epigenetic alterations, can also play a role in conferring resistance to LGK974 treatment.

The activity of WNT signalling depends on the accumulation and translocation of β-catenin to the nucleus, one of the hallmarks for the initiation of tumourigenesis in a variety of human cancers, including CRC[23]. We find that pharmacological blockade of WNT ligand secretion resulted in translocation of β-catenin from the cytoplasm and nucleus to the plasma membrane, decreased β-catenin dependent Tcf/LEF transcriptional activity, and cell growth impairment despite massive molecular evolution of resistant derivatives.

Metastatic CRC patients with BRAF mutant tumours are characterised by poor response rates to the anti-EGFR monoclonal antibodies (moAb) panitumumab and cetuximab and poor prognosis, with a median overall survival of only about 9 to 12 months. Despite important clinical benefit recently achieved by combinatorial treatment with BRAF, EGFR and MEK inhibitors, clinical responses are short-lived due to acquisition of secondary resistance. Preclinical and clinical findings unveiled molecularly heterogeneous mechanisms by which cells evade BRAF targeted therapies[3,48–51], that in turn calls for subsequent rounds of therapy, based on the novel molecular landscape acquired. Recent studies highlighted co-occurrence of genetic alterations in RNF43 and BRAF in CRCs[52], identifying a subset of patients with putative selective sensitivity to pharmacological blockade of the WNT pathway.

Indeed, we observed that a BRAF-mutated patient tumour, which rapidly developed secondary resistance to dual blockade of MAPK pathway, retained strong WNT pathway dependency. Here, interference with the activity of WNT ligands limited tumour growth both in vitro (patient-derived organoids) and in vivo (xenopatient).

Comprehensive analyses of CRCs carried out by the Cancer Genome Atlas consortium highlighted that molecular changes lead to deregulation of four main signalling routes including TP53, TGF-beta, WNT and the Receptor Tyrosine Kinase (RTK)-RAS pathway[20]. Blockade of oncogenic receptor tyrosine kinases in advanced CRC patients is hampered by intrinsic and acquired resistance, even when vertical combinations of inhibitors (for instance EGFR, BRAF and MEK triplet combinatorial regimens) are applied[31]. Recent preclinical data indicate that acquired resistance to WNT pathway modulation by the porcupine inhibitor LGK974 can also emerge[35]. The functional consequences of simultaneous targeting distinct signalling nodes known to be deregulated in colorectal tumours are much less investigated.

Interference with ancestral WNT pathway mutations per se, although effective, does not exert prolonged control of tumour growth (both in vitro and in vivo) and horizontal inhibition of MAPK and WNT pathways was not effective when resistance to MAPK was already established. On the contrary, we found that

 

dual blockade *ab initio* (before onset of heterogeneous mechanisms of resistance) led to a strong and durable effect and can be therefore exploited to restrain clonal evolution, and prevent onset of resistance. This suggests that addition of WNT inhibitors to clinically approved kinase inhibitors might provide long-term clinical benefits for CRC patients.

The peculiar oncogene dependence of CRC perhaps reflects the requirement of normal colonic tissue for high WNT activity, retained when cells transform and remaining in place even after profound genomic and biological drifts associated with development of drug resistance. Whether this phenomenon could be observed in other tumour types request further investigations.

In summary the remarkable dependency of CRCs upon ancestral oncogenic alterations offers the rationale for the development of novel cancer therapies and combinatorial strategies designed to suppress, or even prevent, the emergence of resistance in colorectal tumours.

## Methods

**Cell culture and generation of resistant CRC cells.** All cell lines were maintained in their original culturing conditions according to supplier guidelines. Cells were ordinarily supplemented with FBS at different concentrations, 2mM L-glutamine, antibiotics (100 U/mL penicillin and 100 mg/mL streptomycin) and grown in a 37 °C and 5% $CO_2$ air incubator. Cells were routinely screened for absence of Mycoplasma contamination using the Venor® GeM Classic kit (Minerva biolabs). The identity of each cell line was last checked no less than 3 months before performing experiments by PowerPlex® 16 HS System((Promega), throught Short Tandem Repeats (STR) at 16 different loci (D5S818, D13S317, D7S820, D16S539, D21S11, vWA, TH01, TPOX, CSF1PO, D18S51, D3S1358, D8S1179, FGA, Penta D, Penta E and amelogenin). Amplicons from multiplex PCRs were separated by capillary electrophoresis (3730 DNA Analyser, Applied Biosystems) and analysed using GeneMapper v.3.7 software (Life Technologies).

JVE109 CRC cells were obtained by Dr. T. van Wezel, Department of Pathology, Leiden, University Medical Center. Origin of the other parental cell lines was previously published in ref. [53]. BRAF V600E mutant VACO6 and JVE109 resistant derivatives were generated by continuous treatment with dabrafenib (300 nM) alone, combination of dabrafenib and cetuximab (50 μg/mL), LGK974 (250 nM), or combination of dabrafenib and LGK974 until resistant derivatives emerged. HT29 resistant cells were generated by constant treatment with dabrafenib 5 μM and cetuximab 5 μg/mL. All the other resistant cell lines employed in this study have been previously described[2,3,5–7,32].

**Exome analysis of CRC resistant to targeted therapies.** Genomic DNA (gDNA) was extracted using ReliaPrep® gDNA Tissue Miniprep system System (Promega) and sent to IntegraGen SA (Evry, France) that performed library preparation, exome capture, sequencing and data demultiplexing. Final DNA libraries were pair-end sequenced on Illumina HiSeq4000 and FASTQ files produced by IntegraGen were analysed at Candiolo Cancer Institute. Raw data showed a 145× median depth and a 97.5% mean coverage. Data alignment were performed using BWA-mem algorithm[54] on hg38 human reference genome. Resulting files were cleaned of PCR duplicates by "rmdup" samtools command[55]. For each cell line, somatic mutation analysis was performed subtracting variations found in parental (sensitive) sample to resistant counterpart accordingly to what has been previously published[56]. For each resistant cell line, gene copy number (GCN) was computed as follow: first the median read depth of the target regions was calculated; next, for each gene the median read depth was obtained and then divided by the former value.

**Clonal evolution analysis.** Tumour evolution of resistant cell lines was inferred through EXPANDS[57]. This tool estimates tumour cellular prevalence and the number of clonal expansions from nucleotides and gene copy number alterations. EXPANDS results were processed in order to build the trees using the matrix of mutations that inhabit each subpopulation. Clonal evolution has been built as follows: clones containing variations that appear for the first time are defined as father; next, subpopulations containing the same alterations and new ones are assigned to their respective ancestor; and so on, recursively.

**Drug proliferation assays.** CRC cell lines were seeded at different densities (2–5 × $10^3$ cells/well) in medium containing 10% FBS in 96 or 48-well plastic culture plates at day 0. The following day, serial dilutions of the indicated drugs were added to the cells in serum-free medium, while DMSO-only treated cells were included as controls. Plates were incubated at 37 °C in 5% $CO_2$ for indicated time. Cell viability was assessed by measuring ATP content through Cell Titer-Glo® Luminescent Cell Viability assay (Promega). Apoptosis was measured by measuring Caspase 3/7 activity by Caspase-Glo® 3/7 Assay (Promega). Luminescence was measured by TECAN Spark® Plate reader.

**Establishment of primary colorectal cancer and organoids.** Primary colorectal cancer 2D cell lines and 3D organoids were established from tumour tissues obtained from patient's biopsy and patient derived xenografts. Tumour tissues were dissociated into single-cell suspension by mechanical dissociation using the gentleMACS Dissociator (Miltenyi Biotec) and enzymatic degradation of the extracellular matrix using the Tumour Dissociation Kit (Miltenyi Biotec) according to the manufacturer's instructions. The cell suspension was then centrifuged three times at 1200 rpm for 5 min. Supernatants were removed and cell pellets were resuspended with DMEM/F12 medium containing 10% FBS.

To generate 2D primary cell culture, the cell suspensions were passed through a 70-μm cell strainer (Falcon) and resuspended with culture medium DMEM-F12 containing 2 mmol/L L-glutamine, antibiotics (100 U/mL penicillin and 100 μg/mL streptomycin), 50 μg/mL gentamicin, and 10 μmol/L ROCK inhibitor Y-27632 (Selleck Chemicals Inc.) and cultured on collagen-coated dish (Corning) at 37 °C in 5% $CO_2$.

In order to generate tumour-derived 3D organoids, the final cell suspension was centrifuged and washed with PBS twice and the cell pellet was embedded in Basement Membrane Extract (BME; Cultrex BME RGF type 2). Different densities of tumour cells were plated and left to solidify before tumour organoid medium was added and tumour cells were incubated at 37 °C. The composition of Tumour Organoid medium is: DMEM/F12 + Hepes medium supplemented with antibiotics, 1× Primocin (InvivoGen), 1% GlutaMax (Invitrogen), 1× B27 supplement (Invitrogen), 1.25 mM N-acetyl-cysteine (Sigma Aldrich), 10 mM nicotinamide (Sigma Aldrich), 50 ng/mL human EGF (PeproTech), 100 ng/mL R-spondin (R&D), 100 ng/mL Noggin (PeproTech), 10 nM gastrin (Sigma), 500 nM TGFb type I receptor inhibitor A83-01 (Sigma Aldrich), 10 uM p38 MAPK inhibitor SB202190 (Sigma Aldrich) and 10 nM prostaglandin E2 (Tocris). Fresh medium was replaced every 2–3 days. Outgrowing organoids were passaged every 10–15 days after mechanical and enzymatic disruption.

**Droplet digital PCR analysis.** Genomic DNA (gDNA) was extracted using ReliaPrep® gDNA Tissue Miniprep system System (Promega) or Wizard SV96 Genomic DNA Purification System (Promega). Isolated gDNA was amplified using ddPCR Supermix for Probes (Bio-Rad) using KRAS, EGFR and MAP2K1 (PrimePCR ddPCR Mutation Assay, Bio-Rad or custom designed) ddPCR assays for point mutations. ddPCR was then performed according to manufacturer's protocol, and the results were reported as the percentage or fractional abundance of mutant DNA alleles to total (mutant plus wild-type) DNA alleles. Five to ten microliter of DNA template was added to 10 μL of ddPCR Supermix for Probes (Bio-Rad) and 2 μL of the primer and probe mixture. Droplets were generated using the Automated Droplet Generator (Auto-DG, Bio-Rad) where the reaction mix was added together with Droplet Generation Oil for Probes (Bio-Rad). Droplets were then transferred to a 96 well plate and then thermal cycled with the following conditions: 10 min at 95 °C, 40 cycles of 94 °C for 30 s, 55 °C for 1 min followed by 98 °C for 10 min (Ramp Rate 2.5 °C/s). Droplets were analysed with the QX200 Droplet Reader (Bio-Rad) for fluorescent measurement of FAM and HEX probes. Gating was performed based on positive and negative controls, and mutant populations were identified. The ddPCR data were analysed with QuantaSoft analysis software (Bio-Rad) to obtain Fractional Abundance of the mutated DNA alleles in the wild-type or normal background. Fractional Abundance is calculated as follows: F.A.% = (Nmut/(Nmut + Nwt)) × 100), where Nmut is the number of mutant events and Nwt is the number of wild-type events per reaction. The number of positive and negative droplets is used to calculate the concentration of the target and reference DNA sequences and their Poisson-based 95% confidence intervals. ddPCR analysis of normal control DNA (from cell lines) and no DNA template controls were always included. The experiments were repeated at least twice in independent experiments to validate the obtained results.

**Mutational analysis in cell lines.** Genomic DNA samples were extracted by Wizard® SV Genomic DNA Purification System (Promega). For Sanger Sequencing, all samples were subjected to automated sequencing by ABI PRISM 3730 (Applied Biosystems). Primer sequences for ZNRF3 gene are: FW 5′-AGTATGCTCAGCCCTGCCTA-3′; REV 5′- TAGCTGAGGCCCTGGAAGTA-3′.

**Genetic restoration of APC.** CRC cells were detached and seeded in growth medium without antibiotics 18–24 h before electroporation for optimal 70–80% confluence cell density at the time of electroporation. The day after cells were harvested and counted: 1 × $10^6$ cells/mL for each cell line were resuspended in 100 μL Ingenio solution (Ingenio® Electroporation Kits, Mirus) and electroporated with 2 μg of plasmids encoding for wild-type APC or indicated mutant using Amaxa® Nucleofector®. After that, cells were seeded in different 96-well plates in triplicates for multiple readouts. Plates were incubated at 37 °C in 5% $CO_2$ for 48 h. After that, cell viability was assessed by measuring ATP content through Cell Titer-Glo® Luminescent Cell Viability assay (Promega). Cell death was measured by Cell TOX-Green-Cytotoxicity® Assay (Promega). Luminescence and fluorescence were measured by TECAN Spark® Plate reader. Images of Hoechst 3342/Propidium iodide staining were acquired using Cytation3 Imaging Reader® (Biotek) with a 4× objective and analized with ImageJ software. At least two independent experiments were performed for each cell line and condition.

**Immunofluorescence**. Organoids embedded in BME (Cultrex® Basement Membrane Matrix BME) were grown as domes arranged in 8-well chamber slides, in DMEM/F12 10% FBS and treated with LGK974 1 μM. Drug was refreshed every 4 days. After 14 days, organoids were fixed in 4% paraformaldehyde for 30 min at room temperature (RT) and permeabilized with 0.5% Triton-X100 in PBS for 30 min RT. After that, organoids were treated with 1% BSA in PBS for 30 min and incubated overnight at 4 °C with the following primary antibodies diluted in PBS containing 1% of BSA and 1% of donkey serum: mouse monoclonal anti-β-catenin (BD Transduction, CA) or rabbit monoclonal anti-Cleaved Caspase-3 (asp175) (D3E9) (Cell Signalling Technology, USA). After washing, organoids were fluorescently labelled, according to the primary antibody used, with an Alexa Fluor® 555 donkey anti-mouse antibody or Alexa Fluor® 488 donkey anti-rabbit antibody (Molecular Probes, Eugene, USA) diluted 1:400 in PBS containing 1% BSA and donkey serum for 1 h. Nuclei were stained with DAPI. F-actin was stained with Alexa Fluor® 647 Phalloidin (50 μg/mL). Slides were then mounted using the fluorescence mounting medium (Dako, Glostrup, DK) and analysed using a confocal laser scanning microscope (TCS SPE II; Leica, Wetzlar, D).

Cells, grown on glass coverslip, were fixed in 4% paraformaldehyde for 20 min at RT and permeabilized with 0.1% Triton-X100 in PBS for 2 min on ice. Then cells were treated at RT with 1% BSA in PBS for 30 min and incubated for 2 h at RT with the same antibodies and following the same procedures used for organoids.

**Luciferase reporter assay**. CRC cell lines were seeded at $5 \times 10^4$ cells/well in 400 μL growth medium without antibiotics in 24-well plates. The day after, cells were transfected with 0.5 μg of either TOP or FOP expression plasmids using Lipofecta-mine3000 according to the manufacturer's instructions (Life Technologies). Twelve hours after transfection, cells were treated with LGK974 1–2 μM for 24 h prior to luciferase activities being measured using a Glomax-96® Luminometer (Promega). The TOP/FOP ratio was used as a measure of β-catenin driven transcription.

**Q-RT-PCR**. Total RNA was extracted from CRC cells using Maxwell® RSC miRNA Tissue Kit (Promega), according to the manufacturer's protocol. The quantification and quality analysis of RNA was performed by Thermo Scientific Nanodrop 1000 and Bioanalyser 2100 (Agilent). DNA was transcribed using iScript RT Super Mix (BioRad) following the manufacturer's instructions. Q-RT-PCR was performed in triplicate on ABI PRISM 7900HT thermal cycler (Life Technologies) with SYBR green dye. The mRNA expression levels of the AXIN2 and LGR5 genes were normalised to TBP, SDHA and HPRT genes expression. The sequences of the primers (IDT) used for gene expression analyses were: AXIN2 FW 5′-CGGGCATCTCCGGATTC-3′; AXIN2 REV 5′- TCTCCAGGAAAGTTCGGA ACA -3′; LGR5 FW 5′-CAAGCCATGACCTTGGCCCTG-3′; LGR5 REV, 5′-TTTCCCAGGGAGTGGATTCATT -3′; HPRT FW 5′-TCAGGCAGTATAA TCCAAAGATGGT-3′; HPRT REV 5′-AGTCTGGCTTATATCCAACACTTC G-3′; SDHA FW 5′- TGGGAACAAGAGGGCATCTG-3′; SDHA REV 5′- CCAC CACTGCATCAAATTCATG-3′; TBP FW 5′- CACGAACCACGGCACTGAT T -3′; TBP REV 5′- TTTTCTTGCTGCCAGTCTGGAC -3′.

**Western blotting analysis**. Prior to biochemical analysis, all cells were grown in their specific media supplemented with 10% FBS. Indicated cells were treated with 1 μM LGK974 for 24 h or with 100 ng/mL of recombinant human WNT-3a (R&D Systems) for 1 h. After that, total cellular proteins were extracted by solubilizing the cells in EB buffer (50 mM Hepes pH 7.4, 150 mM NaCl, 1% Triton X-100, 10% glycerol, 5 mM EDTA, 2 mM EGTA; all reagents were from Sigma-Aldrich, except for Triton X-100 from Fluka) in the presence of 1 mM sodium orthovanadate, 100 mM sodium fluoride and a mixture of protease inhibitors. Extracts were clarified by centrifugation, normalised with the BCA Protein Assay Reagent kit (Thermo). Western blot detection was performed with enhanced chemilumines-cence system (GE Healthcare) and peroxidase conjugated secondary antibodies (Amersham). The following primary antibodies were used for western blotting (all from Cell Signalling Technology, except where indicated): anti-pLRP6 (Ser1490) (1:1000); anti-LRP6 (C5C7) (1:1000); anti-Axin1 (C76H11) (1:1000); anti-actin (Santa Cruz) (1:3000).

**Patient-derived mouse model**. All animal procedures were approved by the Ethical Committee of the Institute and by the Italian Ministry of Health. The methods were carried out in accordance with the approved guidelines. Tissue biopsy was subcutaneously implanted in 7-week-old NOD-SCID mouse (Charles River Laboratory). After engraftment, the tumour was passaged and expanded until production of two cohorts. The experiments were designed to include the mini-mum amount of mice per group to be scientifically and statistically valid. No statistical methods were used to predetermine sample size. Mice were randomised to an average tumour size of 350–400 mm³. All the animals were included in the randomisation that was considered valid if the differences between the two averages were lower than 10%. There were no data exclusion. Mice were then dosed by oral gavage with vehicle or LGK974 (Catalog No.S7143; Selleck Chemicals) resuspended in 0.5% methylcellulose/0.5% Tween 80 and administered to mice 7.5 mg kg⁻¹ daily. Tumour size was measured twice a week and calculated using the formula: $V = ((d)2 \times (D))/2$ (d = minor tumour axis; D = major tumour axis)

and reported as tumour mass volume (mm³, mean ± CI of individual tumour volume). The investigators were not blinded. The measures were acquired before identification of the cages.

**Statistical analyses**. Statistical significance was determined by unpaired two-tailed Student's *t*-test. $P < 0.05$ was considered statistically significant. Assumption that the data are sampled from populations that follow Gaussian distributions has been tested using the method Kolmogorov and Smirnov.

**Data availability**. Sequencing data generated during our study are available in the European Nucleotide Archive (ENA) with the following accession code PRJEB25113.

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

## Acknowledgements

This study was supported by: European Community's Seventh Framework Programme under grant agreement no. 602901 MErCuRIC (A.B.); H2020 grant agreement no. 635342-2 MoTriColor (A.B. and S.S.); IMI contract n. 115749 CANCER-ID (A.B.); AIRC 2010 Special Program Molecular Clinical Oncology 5 per mille, Project n. 9970 Extension program (A.B. and S.S.); AIRC IG n. 16788 (A.B.); AIRC IG n. 17707 (F.D.N.); AIRC IG 20885 (S.S.); Fondazione Piemontese per la Ricerca sul Cancro-ONLUS 5 per mille 2011 e 2014 Ministero della Salute (A.B.); Fondazione Piemontese per la Ricerca sul Cancro-ONLUS Innovation 5 per mille 2012 MIUR (M.R.); Terapia Molecolare Tumori by Fondazione Oncologia Niguarda Onlus (A.S.B. and S.S.); Genomic-Based Triage for Target Therapy in Colorectal Cancer Ministero della Salute, Project n. NET 02352137 (A.S.B., A.B. and S.S.).

## Author contributions

M.R. and A.B. conceived the study. M.R., S.L., A.L., A.S., B.M., M.M., L.L., S.A., D.O. conducted the experiments and analysed data. M.L. provided CRC cell lines. F.P., A.S.B. and S.S. provided patient samples. G.C. and G.R. performed bioinformatics analysis. M. R., F.D.N. and A.B. wrote the manuscript. A.B. supervised the study.

## Additional information

**Competing interests:** The authors declare no competing interests.

