## [Peer Review File · Nature Communications]

Reviewers' comments:

Reviewer #1 (Remarks to the Author):

This is a potentially interesting report which studies the gene mutation heterogeneity of drug resistant clones. Some information is of value, including the maintenance of ancestral mutations in colorectal cancers, the diverse acquired gene mutations in sub-clones, and that "treatment with targeted therapy, although initially effective, fuels clonal evolution and further amplifies molecular diversity." Such patterns have been observed in different types of cells.

My major concern is its data presentation and unfocused discussion. There are multiple places where further explanations are needed. For example, in the abstract, they proposed to test a hypothesis that interferes with an ancestral oncogenic event shared by all malignant cells that may override heterogenous molecular mechanisms of acquired drug resistance. It is not clear if their studies have supported or contradicted this hypothesis. The authors should spell it out.

Make sure Line 106 ("Loss of function mutations in the APC gene or gain of function mutations in CTNNB1 gene are found in more than 80% sporadic CRCs") is fully supported by current sequencing data.

In line 179, when mentioning "additional mechanisms of drug escape remain to be discovered," the authors should point out that genome chaos, a rapid and massive genome re-organization, represents an important mechanism.

In line 293, it must be noted that drug treatment can also induce gene mutation as well as genome chaos in addition to the selection of low frequency pre-existing mutations.

Line 343, "thus offering broadly applicable options to override heterogeneity," should be tested by the authors (it should not take too much time).

Reviewer #2 (Remarks to the Author):

Russo and colleagues have investigated residual dependency on wnt signalling in colorectal cancers that have been treated with various targeted MAPK pathway agents. Using in vitro and PDX models, they show that despite having evolved well-characterised resistance mutations to various forms of MAPK blockade, and potentially other cryptic genetic changes, the 'MAPK-resistant' cancer cell population shows cell death when normal APC function is restored (or in appropriate cases when wnt-signalling is modulated by administration of a porcupine inhibitor).

The paper is clearly written and the figures are good. The work is a welcome addition to the literature, which in particular builds upon the previous observation that restoration of normal APC function causes regression in mouse and organoid models (Dow – ref 29).

Major comments:

1. Exome sequencing is applied to various cell lines exposed to drug, and exome sequencing of the putative resistant population is analysed with EXPANDs bioinformatics software to infer clonal architectures. The accuracy of the EXPANDs method is questionable: the authors have an opportunity to confirm that the inferred architectures are correct by single cell cloning the resistant pool of cells, and using targeted sequencing (even Sanger sequencing/FISH) to verify co-occurrence/exclusivity of mutations predicted to fall in the distinct clones. The clonality is important to understand the heterogeneity of response (see next point).

B) Exome sequencing is missing for the WIDR and KM12 cell lines assayed in fig 3. Can the clonal architectures be shown in these cases?

C) Relatedly, was there any evidence of (incidental/passenger) wnt pathway mutations post MAPK therapy?

2. Figures 3-5 show various in vitro (& PDX) experiments that demonstrate that restoration of (more physiological) wnt signalling increases cell death despite evolution to MAPK therapy. These data alone do not show that wnt-modulation is effective to control tumour growth, and indeed the longitudinal data in Fig 5d & Fig 6 indicate that the wnt-modulation has a more modest effect. Can the authors show longitudinal cell growth/death data be shown for all cases so that the effectiveness of wnt-modulation be better understood?

B) The next question is then why are (potentially) only some cells killed via modulation of wnt-signalling. Could the authors use sequencing to determine if only one of the clones pre-existing prior to treatment (as identified in fig 2) is selected by wnt-modulation?

3. Fig 6 shows that concomitant blockade of MAPK and wnt-modulation is very effective at controlling untreated cell lines. This is interesting data! In keeping with the rest of the paper, it would be very interesting to see similar experiments (of 'vertical' and 'horizontal' suppression) applied to the MAPK-blockade treated cell lines derived earlier in the manuscript. Are MAPK-blockade resistant cells more effectively treated with a combination therapy or is all MAPK dependence 'lost'? Relatedly, is dual wnt (WT APC restoration) and MAPK treatment effective in classical APC mutant cell lines?

4. The phase "molecular drift" is used to describe the change in cell genetics under treatment. In my mind drift is reserved for situations where there is no selective pressure – might the phrase "molecular evolution" be better?

5. Fig 5d only shows upper confidence intervals – both should be added. (CIs are better than std. error at showing the data range)

Response to Reviewers' comments:

Reviewer #1 (Remarks to the Author):

This is a potentially interesting report which studies the gene mutation heterogeneity of drug resistant clones. Some information is of value, including the maintenance of ancestral mutations in colorectal cancers, the diverse acquired gene mutations in sub-clones, and that “treatment with targeted therapy, although initially effective, fuels clonal evolution and further amplifies molecular diversity.” Such patterns have been observed in different types of cells.

We thank the referee for the supportive statements and for highlighting the implications of our work.

My major concern is its data presentation and unfocused discussion. There are multiple places where further explanations are needed. For example, in the abstract, they proposed to test a hypothesis that interferes with an ancestral oncogenic event shared by all malignant cells that may override heterogeneous molecular mechanisms of acquired drug resistance. It is not clear if their studies have supported or contradicted this hypothesis. The authors should spell it out.

We appreciate the concern of the reviewer; we believe our data openly support the hypothesis that interference with ancestral oncogenic events shared by all malignant cells can override heterogeneous molecular mechanisms of acquired drug resistance. Following the reviewer’s suggestion this concept has been further highlighted in the text.

Make sure Line 106 (“Loss of function mutations in the APC gene or gain of function mutations in CTNNB1 gene are found in more than 80% sporadic CRCs”) is fully supported by current sequencing data.

We thank the referee for this suggestion. Yaeger and colleagues (Yaeger et al., Cancer Cell 2018) recently performed a prospective targeted sequencing of 1,134 CRCs, and found that the most common genomic alterations in this tumor type affect the APC gene (79%). Other recurrently mutated genes in the WNT pathway involve CTNNB1 (8%) and RNF43 (9%), resulting in an overall pathway alteration frequency of 93% in MSI-H tumors and 85% in MSS tumors respectively. These results confirmed the frequency of WNT pathway alterations in colorectal cancers reported by TCGA Network (Network TCGA, Nature 2012). The reference from Yaeger and colleagues has been added to the manuscript.

In line 179, when mentioning “additional mechanisms of drug escape remain to be discovered,” the authors should point out that genome chaos, a rapid and massive genome re-organization, represents an important mechanism.

We thank the referee for this suggestion and have added the concept of genome chaos in the discussion (see also point below).

In line 293, it must be noted that drug treatment can also induce gene mutation as well as genome chaos in addition to the selection of low frequency pre-existing mutations.

We now acknowledged these aspects in the revised discussion as recommended.

Line 343, “thus offering broadly applicable options to override heterogeneity,” should be tested by the authors (it should not take too much time).

A major limitation for effective clinical treatment of cancer patients with targeted agents such as kinase inhibitors is the seemingly unavoidable onset of secondary resistance. We and others have previously shown that secondary resistance to targeted therapies is often associated with heterogeneous molecular mechanisms which cannot be easily eradicated with a single therapeutic strategy (Russo et al., *Canc Discov* 2016; Pietrantonio et al., *Canc Discov* 2016).

As an example, we found that patients who developed clinical resistance to EGFR, BRAF or MEK blockade had highly heterogeneous tumors that could not be controlled with therapies rationally based on the mechanism they had developed, and this was likely due to their pleiotropic mechanisms of resistance (Russo et al *Canc Discov* 2016; Pietrantonio et al., *Canc Discov* 2016).

In line with this, when we treated a panel of CRC cells, who had developed multiple mechanisms of secondary resistance to targeted therapies, with rational combination of oncogenic signaling inhibitors, we observed an heterogeneous pattern of response. As an example in Rebuttal Figure 1, combination of trametinib and cetuximab was effective on DIFI R2 and HCA46 R1 resistant cells, had modest effect on JVE109 R2, VACO6 R2 and R3, and no efficacy on JVE109 R3, VACO R1 and WIDR R3. Furthermore, we observed an enhancement of cell growth in HT29 R1 cells.

To overcome this issue in this manuscript we propose an unconventional approach. We find that interference with trunk-clonal alterations (WNT pathway) led to cell death in all the resistant CRC cells we tested (Rebuttal Figure 2; Fig.4 and Supplementary Figs 2, 4 of the original manuscript) independently of the genetic landscape and survival mechanisms they acquired.

Rebuttal Figure1. The indicated cell lines were treated with the MEK inhibitor trametinib alone or in combination with EGFR monoclonal antibody cetuximab (cetux). After 5 days, cell viability was assayed by ATP assay. Dashed line indicates 40% cell viability.

Rebuttal Figure2. (A) The indicated cell lines were treated with WNT pathway inhibitor LGK974 for 5 days, after that cell viability was measured by luminescent assay. (B) The indicated cell lines were transfected with control electroporation buffer (mock) or with a plasmid expressing WT APC. 48h after, cell viability was assessed by luminescent assay. Dashed line indicates 40% cell viability. Modified from Figure 4A (A) and Supplementary Figure 2 (B) of the original manuscript.

Reviewer #2 (Remarks to the Author):

Russo and colleagues have investigated residual dependency on wnt signalling in colorectal cancers that have been treated with various targeted MAPK pathway agents. Using in vitro and PDX models, they show that despite having evolved well-characterised resistance mutations to various forms of MAPK blockade, and potentially other cryptic genetic changes, the 'MAPK-resistant' cancer cell population shows cell death when normal APC function is restored (or in appropriate cases when wnt-signalling is modulated by administration of a porcupine inhibitor).

The paper is clearly written and the figures are good. The work is a welcome addition to the literature, which in particular builds upon the previous observation that restoration of normal APC function causes regression in mouse and organoid models (Dow – ref 29).

We thank the reviewer for the positive comments and for underscoring the relevance of our findings.

Major comments:

1. Exome sequencing is applied to various cell lines exposed to drug, and exome sequencing of the putative resistant population is analysed with EXPANDs bioinformatics software to infer clonal architectures. The accuracy of the EXPANDs method is questionable: the authors have an opportunity to confirm that the inferred architectures are correct by single cell cloning the resistant pool of cells, and using targeted sequencing (even Sanger sequencing/FISH) the verify co-occurrence/exclusivity of mutations predicted to fall in the distinct clones. The clonality is important to understand the heterogeneity of response (see next point).

We thank the referee for raising the point. Following his/her suggestion we performed single cell cloning of resistant CRC cells in which multiple mechanisms of resistance were identified (HT29 R1, JVE109 R3 and VACO6) to verify co-occurrence/exclusivity of mutations as requested. We

found that VACO6 cells were not conducive for this experiment since they grow in suspension and continuously form aggregates, which impaired effective single cell dilution. Accordingly, we proceeded with single cell dilution (SCD) of HT29 R1 and JVE109 R3 resistant cells.

HT29 R1 cells were single cell diluted in 96wells plates. After 4-5 weeks, we performed ddPCR analysis of individual clones for mutant alleles previously identified in the population (Figure 2 of original manuscript). As reported in Rebuttal Table 1, when we analyzed gDNA from HT29 R1-derived clones, in all but one instance we detected either the EGFR S492R or KRAS E63K mutation as predicted by the clonal architectures inferred by the bioinformatics analysis. In a single case both mutations were detected, although at very different fractional abundance (22.5% of EGFR S492R and 0.95% of KRAS E63K) (Rebuttal Table 2). The most likely explanation is that this well contained a mixed population of cells.

We found that JVE109 R3 cells have poor clonogenic potential. To increase the number of clones we therefore performed several rounds of limiting dilution followed by molecular profiling. ddPCR analysis revealed prevalently clones harboring either the KRAS G12D or the MAP2K1 I103N alteration (Rebuttal Table 1). Also in this case we occasionally found wells containing both mutations (Rebuttal Table 2). The different fractional abundance suggests a possible mixed population of cells, which is supported by the fact that JVE109 tend to form aggregates.

These results indicate that the sub-clonal architecture of resistant cellular population inferred by EXPANDS was correct at least in the instances in which this could be experimental tested.

While these results suggest that EXPANDS can be used to properly predict the clonal structure of cell populations, we share the referee's concern about EXPANDS and other bioinformatics tools and we have therefore added this aspect in the revised discussion of the manuscript.

Sample	Fractional Abundance (%)
HT29 R1	12/66 EGFR S492R
	3/7 KRAS E63K
	1/7 MIXED
JVE109 R3	10/17 KRAS G12D
	4/17 MAP2K1 I103N
	3/17 MIXED

Rebuttal Table 1. Results of ddPCR analysis on the indicated resistant cell models. CRC cells were single cell diluted in 96-wells plate, after 4-5 weeks ddPCR analysis was performed on gDNA extracted from clones.

Sample	HT29 R1-clones	Fractional Abundance of EGFR S492R (%)	Fractional Abundance of KRAS E63K (%)
HT29 R1	Clone 1		19.1
	Clone 2	25.6	
	Clone 3	24.6	
	Clone 4	21.5	
	Clone 5		39.2
	Clone 6		2.9
	Clone 7	22.5	0.95
Sample	JVE109 R3-clones	Fractional Abundance of KRAS G12D (%)	Fractional Abundance of MAP2K1 I103N (%)
JVE109 R3	Clone 1	38	
	Clone 2		34
	Clone 3	57	
	Clone 4	41	
	Clone 5	32	
	Clone 6		32
	Clone 7	55	
	Clone 8	71	
	Clone 9		33
	Clone 10	48	
	Clone 11		24
	Clone 12	19	
	Clone 13	54	
	Clone 14	60	
	Clone 15	43	7.6
	Clone 16	36.6	9.2
	Clone 17	4	34.3

Rebuttal Table 2. Table lists the relative fractional abundance of mutated alleles detected by ddPCR analysis of single clones derived from indicated CRC resistant cell populations.

B) Exome sequencing is missing for the WIDR and KM12 cell lines assayed in fig 3. Can the clonal architectures be shown in these cases?

We had initially limited WES to a subset of the models to limit costs. Following the referee's suggestion we performed additional exome sequences on WIDR and KM12 parental cells and their derivative resistant WIDR R3 and KM12 R1.

WES analysis of KM12 cells reached a depth of 125X and coverage of 98.6%; while depth and coverage of exomes performed on WIDR cells were 140X and 98.6% respectively. As for the cell models already reported in the paper we inferred the sub-clonal architecture of each resistant population taking into account gene copy number, synonymous and non-synonymous somatic alterations acquired following selective pressure of targeted agents.

As reported in rebuttal Figure 3, treatment with combination of vemurafenib and cetuximab (WIDR R3) or NTRK1 entrectinib (KM12 R1) led to concomitant evolution of multiple cellular lineages characterized by defined genetic alterations. The clonal architectures of WIDR R3 and KM12 R1 resistant cells have been added to main Figure 2 as requested.

While performing the additional WES analyses, we repeated bioinformatic analysis of all the exomes reported in Figure 2 of the submitted manuscript. This led us to notice that KRAS A11T mutation identified in JVE109 R3 was present in the same sequencing reads as KRAS G12D, indicating they co-occur on the same allele. While the G12D is a well-established activating variant, the role of KRAS A11T in driving secondary resistance in this model is much less clear. Accordingly we removed this mutation from the phylogenetic tree in Figure 2 and relative tables. We apologize for not noticing this earlier.

Rebuttal Figure 3. The bioinformatics tool EXPANDS was used to infer the clonal architectures of indicated CRC resistant cells using gene copy number, synonymous and non-synonymous somatic. Each circle represents a sub-clonal population; numbers indicate non-synonymous variations acquired during clonal drift. Length of the branches is proportional to the number of variants (synonymous and non-synonymous) acquired by individual clones; ancestral branches define the main color of its sub-clones. Sub-populations carrying somatic alterations known to drive drug resistance are highlighted.

C) Relatedly, was there any evidence of (incidental/passenger) wnt pathway mutations post MAPK therapy?

We thank the referee for raising this point. Intrigued by this suggestion we went back to WES data of MAPK resistant CRC cells and scrutinized WNT pathway mutations associated with acquired resistance. Incidental/passenger WNT pathway mutations at resistance to MAPK inhibition were absent in the four MSS CRC cells (WIDR, HT29, HCA46, DIFI). The only WNT pathway alterations retrieved were in MSI cell line JVE109 and VACO6, possibly as a consequence of higher mutational burden that characterize MSI CRC cells. To identify mutations likely to be ‘driver’ from passengers we assessed their occurrence in the COSMIC database. As reported in rebuttal Table 3, the ZNRF3 R859* variant was the only alteration previously reported in Cosmic.

MAPK-resistant Cell Line	Reported in COSMIC	Gene Name	AA change	Variation Effect	% mutant reads	Gain/Loss
JVE109 R3	3	ZNRF3	p.R859*	stopgain	18,5714	GAIN T
JVE109 R2	0	WNT3	p.G157S	nonsynonymous	16,092	GAIN T
	0	WNT6	p.N339S	nonsynonymous	4,67836	GAIN G
VACO6 R1	0	LRP4	p.L1016M	nonsynonymous	3,54331	GAIN T
VACO6 R2	0	LRP4	p.L1016M	nonsynonymous	9,5941	GAIN T

Rebuttal Table 3. The table shows mutations in WNT pathway identified in CRC cells resistant to different therapeutic regimens. Variants were identified comparing parental and resistant cells. Variants with a fractional abundance below 1% were not considered.

2. Figures 3-5 show various in vitro (& PDX) experiments that demonstrate that restoration of (more physiological) wnt signalling increases cell death despite evolution to MAPK therapy. These data alone do not show that wnt-modulation is effective to control tumour growth, and indeed the longitudinal data in Fig 5d & Fig 6 indicate that the wnt-modulation has a more modest effect. Can the authors show longitudinal cell growth/death data be shown for all cases so that the effectiveness of wnt-modulation be better understood?

We thank the referee for raising this interesting point. To tackle the issue we performed multiple experiments. MAPK-resistant CRC cells (sensitive to pharmacological inhibition of WNT pathway) were treated with the porcupine inhibitor LGK974 in long-term assays. Cell viability and death were analyzed longitudinally for 21 days at indicated time points. As shown in Rebuttal Figure 4, we observed progressive cell growth impairment from day 5 to day 16. Notably, in all cell lines but one (JVE109 R2) we observed a minor increase of cell viability at day 21 suggesting the presence of 'persister' cells that might have survived WNT pathway inhibition.

In line with this, the induction of caspase activity starts to regress after 3 weeks of treatment. These data support our findings that interference with ancestral WNT pathway mutations per se, although effective, does not exert prolonged control of tumor growth (both *in vitro* and *in vivo*). Indeed treatment of CRC cells with porcupine inhibitor inexorably led to onset of resistance (Figure 6 of original manuscript), while concomitant inhibition of both MAPK and WNT pathway *ab initio* prevented onset of resistance (Figure 6 of original manuscript).

Rebuttal Figure 4. MAPK-resistant cells were seeded in 48wells and treated with 1 μ M LGK974. Drug treatment was repeated every week. Cell viability (black line, left y-axis) and cell death (caspase 3/7 activity red line, right y-axis) were analyzed at the indicated time points.

B) The next question is then why are (potentially) only some cells killed via modulation of wnt-signalling. Could the authors use sequencing to determine if only one of the clones pre-existing prior to treatment (as identified in fig 2) is selected by wnt-modulation?

Intrigued by this question we performed additional pharmacological screening on MAPK-resistant cells. We treated CRC resistant cells with WNT inhibitor for 2 weeks and then extracted gDNA from cells that survived to pharmacological treatment.

ddPCR analysis unveiled that the fractional abundance of pre-existing mutant alleles remain substantially unchanged between untreated and LGK974-treated cells over 2 weeks (Rebuttal Figure 5). Note that slight fluctuations of fractional abundance of mutated alleles are detected also in untreated cells (NT) comparing the two biological replicates (upper and lower panel).

We therefore speculate that interference with WNT pathway induces cell death in CRC cells independently from the oncogenic alterations they had acquired. We cannot exclude that other mechanisms, genetic or epigenetic, can allow a small percentage of cells to persist to LGK974 treatment.

Rebuttal Figure 5. MAPK-resistant cells were treated with 1 μ M LGK974 for 2 weeks. After that, gDNA was extracted from control untreated (NT) and LKG974-treated cells. ddPCR analysis was performed to measure the fractional abundance of mutated alleles previously identified as mechanisms of secondary resistance to MAPK inhibition. A and B indicate independent biological replicates of the experiment.

3. Fig 6 shows that concomitant blockade of MAPK and wnt-modulation is very effective at controlling untreated cell lines. This is interesting data! In keeping with the rest of the paper, it would be very interesting to see similar experiments (of 'vertical' and 'horizontal' suppression) applied to the MAPK-blockade treated cell lines derived earlier in the manuscript. Are MAPK-blockade resistant cells more effectively treated with a combination therapy or is all MAPK dependence 'lost'?

The reviewer raises a relevant point. To test the hypothesis we treated BRAF mutant CRC cells made resistant to MAPK inhibitors with LGK974 alone or in combination with the MAPK pathway inhibitor dabrafenib. Notably, MAPK blockade reduced the ability of WNT inhibitor to impair growth in most of the models (Rebuttal Figure 6).

The only exception was the JVE109 R2 in which horizontal inhibition of MAPK and WNT pathways induced a more pronounced inhibition than WNT blockade alone. We note that this is the only resistant population not carrying RAS mutations at resistance (Supplementary Tables 1 and 2 of the original manuscript). We speculate that in all models carrying concomitant BRAF V600E and KRAS

mutations, BRAF V600E inhibition by dabrafenib induced rebound activation of the MAPK pathway therefore counteracting the WNT inhibitor efficacy. This phenomenon (rebound effect) was elegantly described by Lito and colleagues (Cancer Cell 2012) and occurs when KRAS oncogenic mutations emerge in a BRAF mutated background.

Contrary to the lack of efficacy of MAPK in cells that already developed secondary resistance, horizontal inhibition of MAPK and WNT pathway *ab initio* (before onset of heterogeneous mechanisms of resistance) led to a strong and durable effect (Figure 6 of submitted manuscript). The new results have been included in the revised manuscript.

Rebuttal Figure 6. MAPK-resistant CRC cells were treated with LGK974 alone or in combination with MAPK inhibitor dabrafenib (DAB). After 5 days, cell viability was assayed by ATP assay.

Relatedly, is dual wnt (WT APC restoration) and MAPK treatment effective in classical APC mutant cell lines?

Prompted by the referee's question, we tested combination of WT APC restoration and MAPK inhibition in parental APC mutant cells and on their derivatives harboring distinct mechanisms of secondary resistance. As shown in Rebuttal Figure 7, MAPK treatment increases the efficacy of APC restoration in parental (sensitive) lines. Once resistance is established MAPK inhibition does not further enhance the impact of WNT pathway restoration.

Rebuttal Figure 7. APC mutated CRC cells and their MAPK-resistant derivatives were transfected with plasmid expressing intact WT APC or control electroporation buffer (mock). After transfection cells were seeded in 48wells plates with or without cetuximab (for DIFI wt and resistant cells), dabrafenib + cetuximab (for WIDR and HT29 wt and resistant cells). After 48h, cell viability was assessed. Representative graphs of two independent experiments for each cell line are reported.

4. The phase “molecular drift” is used to describe the change in cell genetics under treatment. In my mind drift is reserved for situations where there is no selective pressure – might the phrase “molecular evolution” be better?

We agree with the reviewer, and have modified the text accordingly.

5. Fig 5d only shows upper confidence intervals – both should be added. (CIs are better than std. error at showing the data range)

We thank the referee for this suggestion and have modified Figure 5d accordingly.

REVIEWERS' COMMENTS:

Reviewer #1 (Remarks to the Author):

Now this MS is ready to be accepted.

Reviewer #2 (Remarks to the Author):

The authors have provided a very thorough and convincing response to my previous comments, and I congratulate them on a very nice study.

I note that the majority of the additional data produced in response to my review does not appear to have been added to the revised manuscript, and would strongly encourage the authors to add it. In particular, I would suggest supplementary inclusion of both the ExPANDs validation (Tables R1&2) and the longitudinal birth/death assays (Figure R4).

The data in Figure R5 are really quite surprising (I had assumed differential sensitivity to wnt-modulation between subclones was likely) and so important to highlight. It suggests that the trunkal drivers remain absolutely key!

Might Figure R7 be included as panels in Figure 6?

RESPONSE TO REVIEWERS' COMMENTS:

Reviewer #1 (Remarks to the Author):

Now this MS is ready to be accepted.

We thank the reviewer for the positive decision.

Reviewer #2 (Remarks to the Author):

The authors have provided a very thorough and convincing response to my previous comments, and I congratulate them on a very nice study.

We thank the reviewer for kind comments, and we are glad to have satisfied his/her requests.

I note that the majority of the additional data produced in response to my review does not appear to have been added to the revised manuscript, and would strongly encourage the authors to add it. In particular, I would suggest supplementary inclusion of both the ExPANDs validation (Tables R1&2) and the longitudinal birth/death assays (Figure R4).

Following referee's suggestion, we have now incorporated in the manuscript Tables R1 and 2 (Supplementary Tables 3 and 4); and Figure R4 (Supplementary Fig. 6)

The data in Figure R5 are really quite surprising (I had assumed differential sensitivity to wnt-modulation between subclones was likely) and so important to highlight. It suggests that the trunkal drivers remain absolutely key!

We agree with the reviewer and have added Figure R5 as new Figure 6 in the revised manuscript. Might Figure R7 be included as panels in Figure 6?

We agree with the reviewer and have added the Figure R7 as Figure 7a in the revised manuscript.